# Modeling of the HIV-1 Life Cycle in Productively Infected Cells to Predict Novel Therapeutic Targets

**DOI:** 10.3390/pathogens9040255

**Published:** 2020-03-31

**Authors:** Olga Shcherbatova, Dmitry Grebennikov, Igor Sazonov, Andreas Meyerhans, Gennady Bocharov

**Affiliations:** 1Marchuk Institute of Numerical Mathematics, Russian Academy of Sciences (INM RAS), 119333 Moscow, Russia; olga.orlova@phystech.edu; 2Moscow Center for Fundamental and Applied Mathematics at INM RAS, 119333 Moscow, Russia; 3Institute for Personalized Medicine, Sechenov First Moscow State Medical University, 119991 Moscow, Russia; 4College of Engineering, Swansea University, Bay Campus, Fabian Way, Swansea SA1 8EN, UK; i.sazonov@swansea.ac.uk; 5Infection Biology Laboratory, Department of Experimental and Health Sciences, Universitat Pompeu Fabra, 08003 Barcelona, Spain; andreas.meyerhans@upf.edu; 6Institució Catalana de Recerca i Estudis Avançats (ICREA), 08010 Barcelona, Spain

**Keywords:** HIV-1, intracellular replication, mathematical model, sensitivity analysis, antiviral targets

## Abstract

There are many studies that model the within-host population dynamics of Human Immunodeficiency Virus Type 1 (HIV-1) infection. However, the within-infected-cell replication of HIV-1 remains to be not comprehensively addressed. There exist rather few quantitative models describing the regulation of the HIV-1 life cycle at the intracellular level. In treatment of HIV-1 infection, there remain issues related to side-effects and drug-resistance that require further search “...for new and better drugs, ideally targeting multiple independent steps in the HIV-1 replication cycle” (as highlighted recently by Tedbury & Freed, The Future of HIV-1 Therapeutics, 2015). High-resolution mathematical models of HIV-1 growth in infected cells provide an additional analytical tool in identifying novel drug targets. We formulate a high-dimensional model describing the biochemical reactions underlying the replication of HIV-1 in target cells. The model considers a nonlinear regulation of the transcription of HIV-1 mediated by Tat and the Rev-dependent transport of fully spliced and singly spliced transcripts from the nucleus to the cytoplasm. The model is calibrated using available information on the kinetics of various stages of HIV-1 replication. The sensitivity analysis of the model is performed to rank the biochemical processes of HIV-1 replication with respect to their impact on the net production of virions by one actively infected cell. The ranking of the sensitivity factors provides a quantitative basis for identifying novel targets for antiviral therapy. Our analysis suggests that HIV-1 assembly depending on Gag and Tat-Rev regulation of transcription and mRNA distribution present two most critical stages in HIV-1 replication that can be targeted to effectively control virus production. These processes are not covered by current antiretroviral treatments.

## 1. Introduction

The Human Immunodeficiency Virus Type 1 (HIV-1) continues to threaten global health and represents a significant burden for health-care systems. Intensive research on HIV and its induced acquired immunodeficiency syndrome (AIDS) for more than 35 years has uncovered unprecedented details of virus and virus-host biology. However, it remains unclear how to cure individuals that are already infected. Main reasons for this unresolved issue are the extraordinary diversity of HIV and its capacity to remain latent within infected cells and thus escape immune surveillance and the action of antiviral drugs.

The complex, multifactorial and nonlinear behavior of the host response to HIV-1 necessitates the application of mathematical modeling for analysis and description of individual’s infection dynamics and for predicting its sensitivity to various drugs. The interventions and conditions that need to be explored in various combinations include (i) antiretroviral drugs; (ii) immune-based therapies; and (iii) cell motility-affecting cytokines. We have recently applied mathematical modeling to predict the impact of PD-L1 blockade on viral load and CD4 T cell gain in HIV-infected individuals [1]. The requirements for cytotoxic CD8 T cells in terms of their frequency and spatial dynamics within lymphoid tissues have been examined in another study [2].

A major progress in controlling HIV-1 infection is related to highly active antiretroviral therapy (HAART). Many drugs have been evaluated until today, targeting six distinct steps in the HIV-1 life cycle, and about 30 drugs can be used in combination [3]. Prevention and control of drug toxicities is one of the key issues in the current HIV-1 management. As the number of elementary biochemical reaction steps in the replication cycle of HIV-1 is much larger than six, there should be other potential targets for ART. This can be explored by formulating a high-resolution model of the intracellular life cycle of HIV-1 and the computational analysis of the sensitivity of virus production to the variation of model parameters that determine the reaction rates.

Since the 1980s, the time when HIV-1 was discovered, several mathematical models of HIV-1 replication in target cells have been built to understand the kinetic determinants of the replication cycle. Among the first models of intracellular HIV-1 kinetics is the model formulated via a system of ordinary differential equations (ODEs) [4]. It was calibrated to describe the growth of HIV-1 in CD4 T lymphocytes. The model considered reverse transcription, nuclear import and DNA integration, transcription, mRNA splicing and export to the cytoplasm, translation, post-translational modifications and transport to the membrane, budding and assembly. The model was used to predict the sensitivity of the virus growth to the reduction of concentrations of doubly spliced, singly spliced and full-length mRNAs, protease, Rev, Tat and reverse transcriptase (RT). The mechanistic model of HIV-1 growth proposed in [5] considered a subset of the HIV-1 life cycle processes ranging from integration of proviral DNA to nuclear export of incompletely spliced mRNA. The study focused on examining the effect of positive and negative feedback loops by Tat and Rev, respectively, on the robustness of virus growth to perturbations resulting from variations of cellular host factors and virus mutations. Subsequently, the nonlinear Tat-Rev regulation was examined as a generator of oscillatory dynamics in the synthesis of viral components [6]. A two-scale model of virus growth, coupling intracellular and cell population levels developed in [7], considered a subset of intracellular stages of HIV-1 replication from random binding to integration of proviral DNA into the host genome. However, it was not used to make novel predictions for HIV-1 growth. The regulation of HIV-1 virion maturation was modeled in [8]. The reaction kinetics of proteolytic processing by HIV-1 protease was developed and calibrated, taking into account all cleavage sites within Gag and Gag-Pol, intermediate products and enzyme forms, enzyme dimerization, the initial auto-cleavage of full-length Gag-Pol as well as self-cleavage of protease. The model was used to identify the rate limiting steps of virion maturation and the parameters with the strongest effects on maturation kinetics. A model of the HIV-1 replication cycle from reverse transcription to translation was developed in [9]. In that model, the infected cell undergoes stochastic transitions between different states corresponding to the stages of virus replication. The kinetics of biochemical species, representing the products of sequential replication stages are modeled using both the mono-compartmental description and the spatially resolved approach. The later considered the infected cell as a circle (2D) and described the movement and interaction of the viral components following an agent-based framework. The model was used to predict the dependence of viral mRNA produced per cell as a function of the multiplicity of infection. A data-driven model of the kinetics of HIV-1 replication intermediates in a T cell line was formulated in [10]. The model described the kinetics of the stages in a non-mechanistic way, i.e., using gamma distribution functions for the net rates of the replication markers. The parameters of the functions were calibrated using original in vitro data.

The primary aim of our study here was to formulate a high-resolution deterministic model of the HIV-1 replication cycle and to calibrate the model using available data on the kinetics and process parameters. The second purpose of the work was to use the model for predicting novel targets of intervention of HIV-1 production via sensitivity analysis of the model. To this end, we applied the variational approach based on adjoint equations.

## 2. Results

### 2.1. Mathematical Model

In this section, we formulate the equations of the mathematical model describing the sequence of biochemical reactions underlying the intracellular replication of HIV-1 in productively infected cells. The schematic representation is shown in Figure 1 modified from [3]. We consider the following replication stages: entry, reverse transcription, integration, transcription, translation, assembly, budding and maturation.

#### 2.1.1. Entry

The entry stage is split into three steps represented in Figure 2:binding of virion to CD4 receptors (glycoprotein gp120 binds to CD4 receptors on the T cell surface),binding to the co-receptor (CCR5 or CXCR4),fusion, i.e., the nucleocapsid is uncoated and the viral RNA is injected into the cell.

The binding of the virion to CD4 is described by equations describing the rates of changes of free- and receptors-bound virion:(1)$$\frac{d[V_{free}]}{dt}=-k_{bind}\left[V_{free}\right]-d\left[V_{free}\right]$$
(2)$$\frac{d[V_{bound}]}{dt}=k_{bind}\left[V_{free}\right]-\left(k_{fuse}+d_{bound}\right)\left[V_{bound}\right]$$
where $\left[V_{free}\right]$ is the number of free virions outside the cell, $\left[V_{bound}\right]$ is the number of virions bound to CD4 and the co-receptor. The respective parameters of the model are described in Table 1.

#### 2.1.2. Reverse Transcription

The fusion of bound virions with the host cell membrane results in the release of the content of the virion into the cell cytoplasm, starting the phase of reverse transcription [11]. During reverse transcription, a double-stranded DNA is created from two single-stranded RNA genomes. According to the scheme presented in Figure 3, the reverse transcription is modeled as three sequential processes:synthesis of minus-strand DNA from viral RNA,synthesis of plus-strand DNA,double-strand DNA formation.

The rates of change of the amount of RNA and proviral DNA species are described by equations:(3)$$\frac{d[RNA_{cor}]}{dt}=k_{fuse}\left[V_{bound}\right]-\left(k_{RT}+d_{RNA_{cor}}\right)\left[RNA_{cor}\right]$$
(4)$$\frac{d[DNA_{cor}]}{dt}=k_{RT}\left[RNA_{cor}\right]-\left(k_{DNA_{t}}+d_{DNA_{cor}}\right)\left[DNA_{cor}\right]$$
where $\left[RNA_{cor}\right]$ is the number of genomic RNA molecules in cytoplasm, $\left[DNA_{cor}\right]$ is the number of proviral DNA molecules synthesized by reverse transcription. The respective parameters of the above equations are described in Table 1.

#### 2.1.3. Integration

After the proviral DNA is synthesized, it associates with virus-encoded integrase (IN) and other proteins as a high-molecular-weight nucleoprotein complex (pre-integration complex, PIC) that is transported into the nucleus for subsequent integration [12]. Integration is the process of viral DNA insertion into chromosomal DNA of the host cell. The viral DNA can also undergo several circularization reactions thus losing the capability to support subsequent replication [12]. The change in the number of viral DNA in the nucleus and the number of integrated DNA are modeled with the following equations:(5)$$\frac{d[DNA_{nuc}]}{dt}=k_{DNA_{t}}\left[DNA_{cor}\right]-\left(k_{int}+d_{DNA_{nuc}}\right)\left[DNA_{nuc}\right]$$
(6)$$\frac{d[DNA_{int}]}{dt}=k_{int}\left[DNA_{nuc}\right]-d_{DNA_{int}}\left[DNA_{int}\right]$$
where $\left[DNA_{nuc}\right]$ is the number of DNA molecules in the nucleus, $\left[DNA_{int}\right]$ stands for the number of integrated DNA.

#### 2.1.4. Transcription

HIV transcription starts when the host cell receives activation signals. It is a process of messenger RNA (mRNA) synthesis. There are three types of mRNA species: full-length (around 9 kb), singly spliced (around 4 kb), doubly spliced (around 2 kb) [3]. After transcription, mRNAs are transported to the cell cytoplasm. There is a temporal regulation of transcription and mRNA distribution by viral Tat and Rev proteins. To describe these stages, we used the scheme in Figure 4 and the parameterization of the feedback regulation similar to [5,6], as specified below:
(7)$$\frac{d[mRNA_{g}]}{dt}=TR\left[DNA_{int}\right]-\left(k_{eRNA_{g}}f_{Rev}+k_{ssRNA_{g}}\left(1-\beta f_{Rev}\right)+d_{RNA_{g}}\right)\left[mRNA_{g}\right]$$
(8)$$\frac{d[mRNA_{ss}]}{dt}=\left(1-\beta f_{Rev}\right)k_{ssRNA_{g}}\left[mRNA_{g}\right]-\left(k_{eRNA_{ss}}f_{Rev}+d_{RNA_{ss}}+k_{dsRNA_{ss}}\left(1-\beta f_{Rev}\right)\right)\left[mRNA_{ss}\right]$$
(9)$$\frac{\left[mRNA_{ds}\right]}{dt}=\left(1-\beta f_{Rev}\right)k_{dsRNA_{ss}}\left[mRNA_{ss}\right]-\left(d_{RNA_{ds}}+k_{eRNA_{ds}}\right)\left[mRNA_{ds}\right]$$
(10)$$\frac{d[mRNAc_{g}]}{dt}=f_{Rev}k_{eRNA_{g}}\left[mRNA_{g}\right]-\left(k_{tp,RNA}+d_{RNA_{g}}\right)\left[mRNAc_{g}\right]$$
(11)$$\frac{d[mRNAc_{ss}]}{dt}=f_{Rev}k_{eRNA_{ss}}\left[mRNA_{ss}\right]-d_{RNA_{ss}}\left[mRNAc_{ss}\right]$$
(12)$$\frac{d[mRNAc_{ds}]}{dt}=k_{eRNA_{ds}}\left[mRNA_{ds}\right]-d_{RNA_{ds}}\left[mRNAc_{ds}\right]$$
where $\left[mRNA_{i}\right]$ is the number of $mRNA_{i}$ molecules in the nucleus and $\left[mRNAc_{i}\right]$ is the number of $mRNA_{i}$ molecules in the cytoplasm, where i∈{g(genomicorfull-length),ss(singlyspliced),ds(doublyspliced)}. The Tat-Rev regulation is parameterized by the following functions
(13)$$TR=TR_{cell}+f_{Tat}\cdot TR_{Tat},$$
and the Michaelis–Menten type equations are used for
(14)$$f_{Tat}=\frac{\left[P_{Tat}\right]}{\theta_{Tat}+\left[P_{Tat}\right]},\;f_{Rev}=\frac{\left[P_{Rev}\right]}{\theta_{Rev}+\left[P_{Rev}\right]},$$
with $\theta_{Tat}={\left(K^{Tat}\right)}^{-1}$, and $\theta_{Rev}={\left(K^{Rev}\right)}^{-1}$ (see [4,5]), respectively. The parameters of the above equations are described in Table 1.

#### 2.1.5. Translation

The viral mRNAs are decoded by ribosomes to produce specific proteins. The proteins then fold into active proteins. The full-length mRNA codes for Gag and Gag-Pol proteins. The singly spliced mRNAs code for gp160, Vif, Vpu and Vpr proteins. The doubly spliced mRNAs code for Nef, Tat and Rev. In our model we account for the kinetics of Gag-Pol, Gag, gp160, Tat and Rev proteins. Their turnover is described by the following set of equations:(15)$$\frac{d[P_{Tat}]}{dt}=k_{trans}f_{ds,Tat}\left[mRNAc_{ds}\right]-d_{p,Tat}\left[P_{Tat}\right]$$
(16)$$\frac{d[P_{Rev}]}{dt}=k_{trans}f_{ds,Rev}\left[mRNAc_{ds}\right]-d_{p,Rev}\left[P_{Rev}\right]$$
(17)$$\frac{d[P_{Gag-Pol}]}{dt}=k_{trans}f_{g,Gag-Pol}\left[mRNAc_{g}\right]-\left(k_{tp,Gag-Pol}+d_{p,Gag-Pol}\right)\left[P_{Gag-Pol}\right]$$
(18)$$\frac{d[P_{Gag}]}{dt}=k_{trans}f_{g,Gag}\left[mRNAc_{g}\right]-\left(k_{tp,Gag}+d_{p,Gag}\right)\left[P_{Gag}\right],$$
(19)$$\frac{d[P_{gp160}]}{dt}=k_{trans}f_{ss,gp160}\left[mRNAc_{ss}\right]-\left(k_{tp,gp160}+d_{p,gp160}\right)\left[P_{gp160}\right]$$
where $\left[P_{j}\right]$ is the number of protein molecules *j*, j∈{Gag-Pol,Gag,gp160,Tat,Rev} and $f_{i,j}$ are the fractions of $mRNA_{i}$ coding $\left[P_{j}\right]$, i∈{g,ss,ds}. The respective parameters of the model are described in Table 1.

#### 2.1.6. Assembly, Budding and Maturation

The late phase of the HIV-1 replication cycle includes the trafficking of the regulatory and accessory proteins and viral glycoproteins to the plasma membrane, the assembly of Gag and Gag-Pol proteins at the plasma membrane with a subsequent encapsidation of the viral RNA genomes, budding of the new virions and their maturation [13]. The viral proteins Gag-Pol, Gag, gp160 undergo several post-translational modifications such as folding, oligomerization, glycosylation, and phosphorylation [4]. After that, the proteins and the full-length mRNA molecules are transported to the membrane where they associate (pre-virion complex) and combine to generate a new virion. The scheme of this late phase of virus replication is shown in Figure 5.

The transport processes to the membrane are modeled by the following kinetic equations (Δ≡{Gag-Pol,Gag,gp160}):(20)$$\frac{d[P_{mem,Gag-Pol}]}{dt}=k_{tp,Gag-Pol}\left[P_{Gag-Pol}\right]-k_{comb}N_{Gag-Pol}\left[RNA_{mem}\right]\prod_{j\in \Delta }\left[P_{mem,j}\right]-d_{mem,Gag-Pol}\left[P_{mem,Gag-Pol}\right]$$
(21)$$\frac{d[P_{mem,Gag}]}{dt}=k_{tp,Gag}\left[P_{Gag}\right]-k_{comb}N_{Gag}\left[RNA_{mem}\right]\prod_{j\in \Delta }\left[P_{mem,j}\right]-d_{mem,Gag}\left[P_{mem,Gag}\right]$$
(22)$$\frac{d[P_{mem,gp160}]}{dt}=k_{tp,gp160}\left[P_{gp160}\right]-k_{comb}N_{gp160}\left[RNA_{mem}\right]\prod_{j\in \Delta }\left[P_{mem,j}\right]-d_{mem,gp160}\left[P_{mem,gp160}\right]$$
(23)$$\frac{d[RNA_{mem}]}{dt}=k_{tp,RNA}\left[mRNAc_{g}\right]-k_{comb}N_{RNA}\left[RNA_{mem}\right]\prod_{j\in \Delta }\left[P_{mem,j}\right]-d_{RNA_{g}}\left[RNA_{mem}\right]$$

The membrane thus encloses viral RNAs with the proteins Gag, Gag-Pol, Vif, Nef, and Vpr while gp160 is anchored in the membrane. The assembly of the pre-virion complex then leads to virion budding. The initially immature budded virions subsequently mature and become infectious for other cells. These last stages are described by equations:(24)$$\frac{d[V_{pre-virion}]}{dt}=k_{comb}\left[RNA_{mem}\right]\prod_{j\in \Delta }\left[P_{mem,j}\right]-\left(k_{bud}+d_{comb}\right)\left[V_{pre-virion}\right]$$
(25)$$\frac{d[V_{bud}]}{dt}=k_{bud}\left[V_{pre-virion}\right]-\left(k_{mat}+d_{bud}\right)\left[V_{bud}\right]$$
(26)$$\frac{d[V_{mat}]}{dt}=k_{mat}\left[V_{bud}\right]-d\left[V_{mat}\right]$$
where $\left[V_{pre-virion}\right]$ is the number of virions on the membrane, $\left[V_{bud}\right]$ is the number of free viruses after budding from the cell and $\left[V_{mat}\right]$ is the number of mature virions outside the cell.

### 2.2. Model Parameters

The model was calibrated using available information on the kinetics and process parameters presented in [4,5,6,9,14,15,16,17,18,19,20,21]. The estimated values and admissible ranges of the model parameters are summarized in Table 1. The variation of threshold parameter $\theta_{Tat}$ results in a temporal shift of the overall kinetics (increase of $\theta_{Tat}$ increases the delay before virion release), while the value of $\theta_{Rev}$ positively influences the rate of virion release. The combination of $\theta_{Tat}$ and $\theta_{Rev}$ parameters influences the overall dynamics in nonlinear way, therefore, they were tuned manually to achieve the expected temporal kinetics of replication cycle stages [10] and physiological levels of transcripts, proteins and mature virions. The initial values of low number of infectious virions ($V_{free}\left(0\right)\leq 5$ virions) result in the integration of $V_{int}\leq 2$ proviral DNA that resembles in vivo scenarios [22] rather than experimental setups with highly susceptible cell lines and high multiplicity of infection. The overall dynamics of the individual components of the HIV-1 replication cycle in activated CD4+ T cells infected with $V_{free}\left(0\right)=4$ infectious virions through their life span of about 36 h [23] is presented in Figure 6. We note from the solution of the calibrated model that the number of membrane anchored Gag molecules is a limiting factor for the assembly of pre-virion complexes since all Gag molecules are being incorporated into the complexes while the abundant levels of other proteins and genomic RNA remain at the membrane.

## 3. Sensitivity Analysis

The calibrated model provides a tool for predicting the dependence of the HIV-1 production by an infected cell to variations of the rates of underlying biochemical processes. The prediction is based on the sensitivity analysis of the model solutions or some functions depending on them [45]. Let us characterize the net outcome of the virus replication cycle by the total number of released virions, $J\left(y\right)=\int_{0}^{T}V_{mat}dt$. Here $\hat{y}\equiv y\left(t,\hat{p}\right)\in R_{+}^{N}$ is the reference (unperturbed) solution to the following initial value problem:(27)$$\begin{array}{c} \frac{d}{dt}y\left(t\right)=f\left(t,y\left(t\right),p\right),\;t\in \left[t_{0},T\right]\\ y\left(t\right)=\phi \left(t,p\right),\;t=t_{0} \end{array}$$

$p\equiv \left[p_{1},p_{2},\ldots ,p_{L}\right]\in R_{+}^{L}$ is a parameter vector. Let the model parameters be changed by small $\delta p=\hat{p}-p$. The local sensitivity of the functional J(y) with respect to the model parameters can be computed using the following system of equations
$$\frac{d}{dt}\left(\frac{\partial J}{\partial p_{i}}\right)=\langle w\left(t\right),\frac{\partial f(t)}{\partial p_{i}}\delta p_{i}\rangle ,\;t\in \left[0,T\right],\;\frac{\partial J}{\partial p_{i}}{|}_{0}=0.$$

The function w(t)≡w(t,p) is the solution to the adjoint problem
(28)$$\begin{array}{c} -\frac{dw(t)}{dt}-{\left[\frac{\partial f}{\partial y}\right]}^{T}w=e_{1},\;t\in \left[t_{0},T\right]\\ w=0,\;t=T \end{array}$$
with
$$e_{1}={\left(0,0,\ldots 0,1\right)}^{T}.$$

For a replication cycle lasting T=36 h [10], and for initial dose of $V_{free}\left(0\right)=4$ virions, the calculated J(p)=1811 virions. The values of the $\frac{dJ}{dp_{i}}$ normalized with respect to parameter values are presented in Figure 7. The left one ranks the sensitivity coefficients for the model parameters which negatively impact the net HIV-1 production when their values are increased. The right bar plot ranks the sensitivity coefficients for the parameters which positively impact the net virus production with their increasing values. The summary of the most critical control parameters characterized by the normalized sensitivity values larger than 1000 and larger than 100, are summarized in Table 2. By comparing the sensitivity analysis data with the existing targets of antiretroviral therapy [3] specified in Figure 8, we identify additional processes which could be considered for ART.

Thus, the ranked processes which display the strongest impact on HIV-1 replication are:maximal achievable level of transcription rate that can be induced by Tat;HIV-1 assembly, specifically sensitive to availability and translation rate of Gag molecules;full-length RNA transport to membrane and degradation;transport of pre-integration complex into nucleus and DNA integration;Rev-mediated regulation of splicing rates and export of full-length RNA;reverse transcription (rate of reverse transcription, RNA degradation, DNA degradation);kinetics of membrane-bound pre-virion complexes and virions;binding and fusion of free virions.

## 4. Discussion

We have formulated mathematical model (system of ODEs) which describes the biochemical reactions underlying the replication life cycle of HIV-1 in activated target cells. The model considers nonlinear regulation of the transcription of HIV-1 mediated by Tat and the Rev-dependent transport of fully spliced and singly spliced transcripts from nucleus to the cytoplasm. The model has been calibrated using available information on the kinetics of various stages of HIV-1 replication.

We have performed the sensitivity analysis of the model to rank the biochemical processes of HIV-1 replication with respect to their impact on the net production of virions by one actively infected cell. The ranking of the sensitivity factors provides a quantitative basis for identifying novel targets for antiviral therapy. Our analysis suggests that (1) HIV-1 assembly depending on Gag and (2) Tat-Rev regulation present two critical stages in HIV-1 replication that can be targeted to effectively control virus production. These processes while being considered to be interesting antiviral targets (i.e., [46,47]) are not targeted by the current antiretroviral treatments. The vital role of the Gag polyprotein precursor and the mature Gag proteins has been discussed recently in [13]. However, the efforts in developing inhibitors of Gag function have not yet resulted in efficacious drugs. The biochemistry of Gag-Pol processing for the HIV-1 assembly deserved the attention from the modeling point of view previously [8]. Our study puts the HIV-1 assembly stage into the context of the whole HIV-1 life cycle and identifies it as the most crucial (in terms of sensitivity) process.

The formulated model and its reduced versions can be used as building blocks for multiscale hybrid models of HIV-1 infection [48,49]. The current version of our model has the limitation that is does not consider the discrete nature and stochasticity of the biochemistry of the infection. This is inherent to the model set up as we considered a deterministic description for it. From the biological point of view, the model lacks a description of intracellular antiviral defense mechanisms and co-factors that facilitate virus propagation. Likewise, we did not incorporate error-prone virus replication that would lead to quasi-species distributions. All these factors can be incorporated to address specific issues that may arise including optimization of antiviral drug combinations and immune-based therapies. Further extensions of the model will require the analysis of stochastic effects to understand the link between the variability across the biochemical parameter values and the heterogeneity of virus production including the response to therapeutic perturbations.

## Figures and Tables

**Figure 1 pathogens-09-00255-f001:**
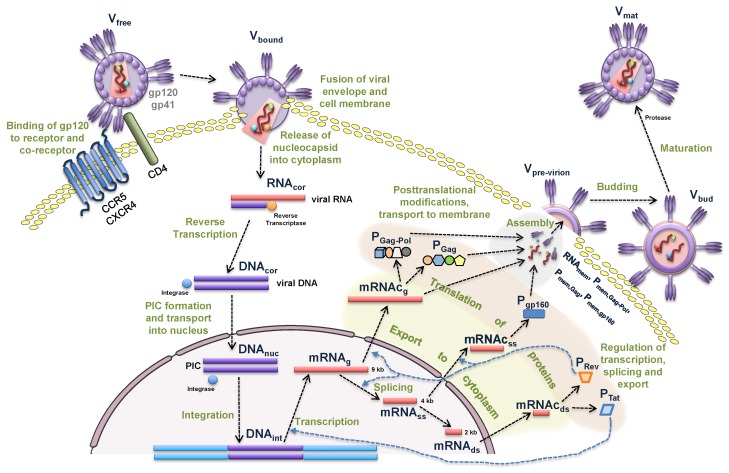
Biochemical scheme of the HIV-1 replication cycle.

**Figure 2 pathogens-09-00255-f002:**
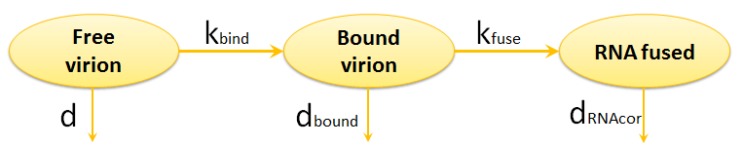
Scheme of HIV-1 entry into the host cell.

**Figure 3 pathogens-09-00255-f003:**
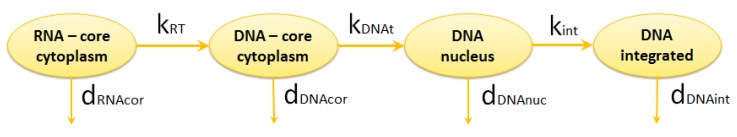
HIV-1 reverse transcription and integration.

**Figure 4 pathogens-09-00255-f004:**
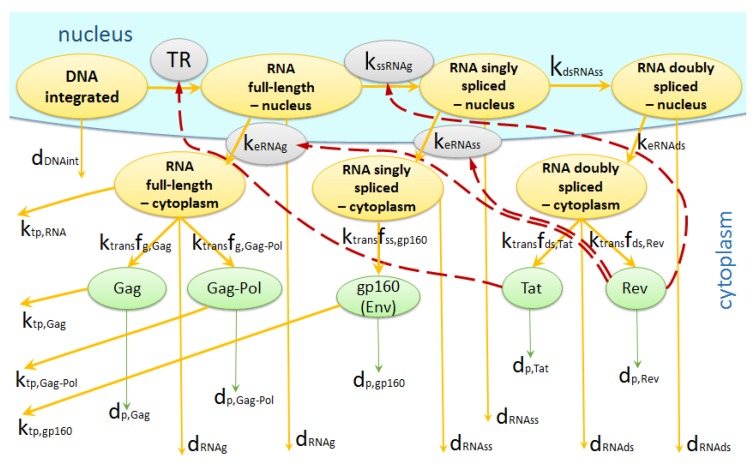
Biochemical events underlying transcription, splicing, export and translation of HIV-1.

**Figure 5 pathogens-09-00255-f005:**
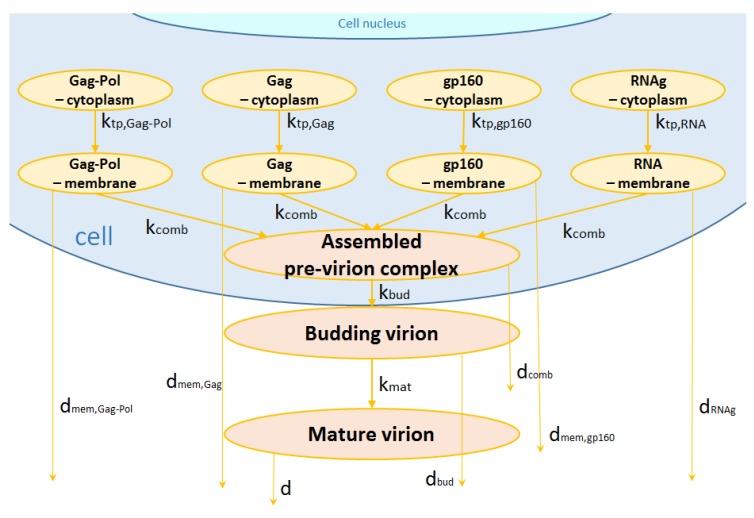
Assembly, budding and maturation.

**Figure 6 pathogens-09-00255-f006:**
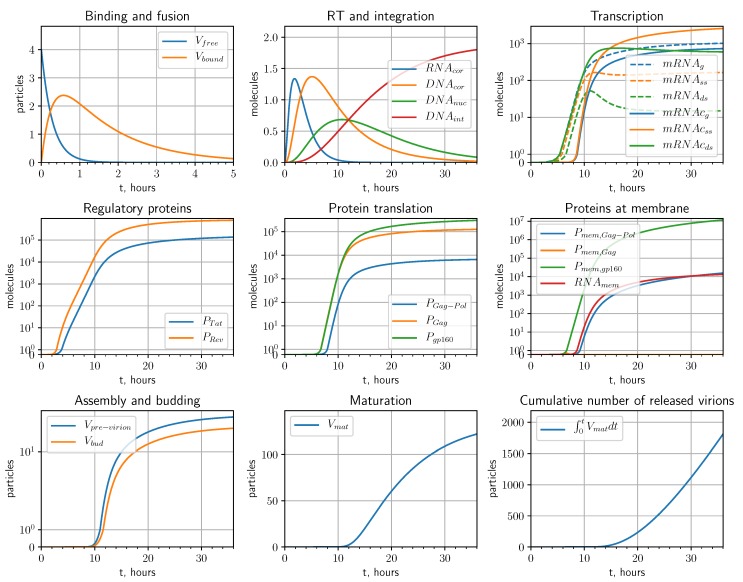
Numerical solution of the calibrated model for the parameter values given in Table 1.

**Figure 7 pathogens-09-00255-f007:**
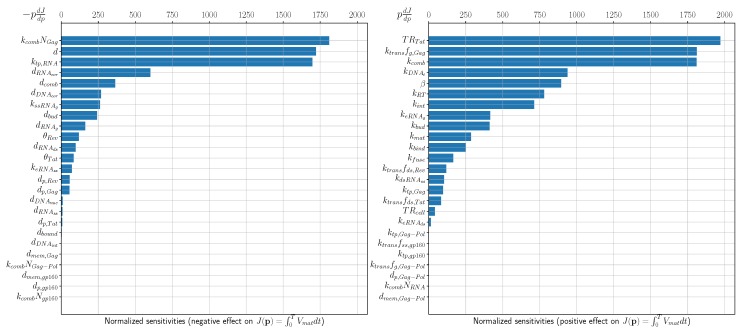
The normalized sensitivity of the functional J(p) to the parameters which have (**left**) negative effect and (**right**) positive effect on HIV-1 production.

**Figure 8 pathogens-09-00255-f008:**
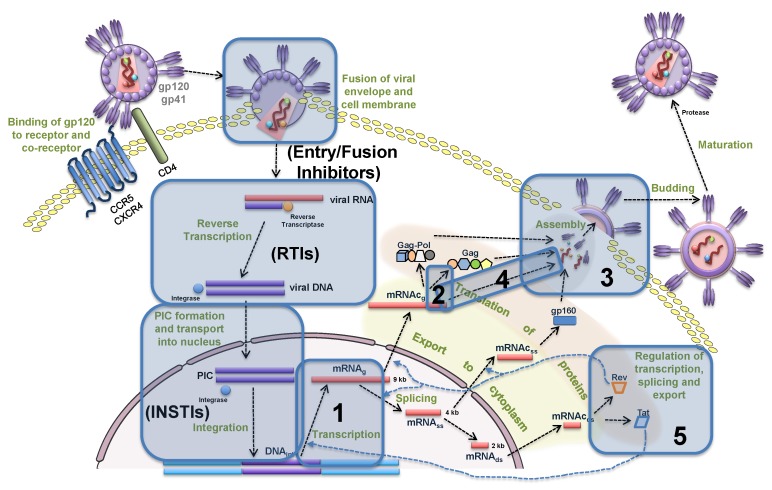
Intracellular replication of HIV-1 with the identified processes showing the strongest impact on virion production. Blue boxes indicate the stages which most strongly affect the virus production. The sets indicated in boxes 1, 2, 3, 4, 5 are the processes with normalized sensitivity values larger than 500 for which effective inhibitors do not exist. The numbers refer to the sensitivity strength with 1 referring to the more strong and 5 to the less strong sensitivity.

**Table 1 pathogens-09-00255-t001:** Estimates of the Calibrated Model Parameters.

Parameter	Description	Value	Range, Relev. Refs.
$k_{bind}$	rate of virion binding to CD4+ T cell membrane	$3.1\;h^{-1}$	(2.1,6.0) [14,24,25]
*d*	clearance rate of free mature virions	0.38 $h^{-1}$	(0.38,1.5) [23,26]
$d_{bound}$	degradation rate of bound virions	$0.0008\;h^{-1}$	$\left(4.8\times 10^{-4},1.9\times 10^{-3}\right)$
			[27,28]
$k_{fuse}$	rate of virion fusion with the cell	0.7 $h^{-1}$	(0.42,2.2) [15,25,29]
$k_{RT}$	reverse transcription rate	0.43 $h^{-1}$	(0.43,1.02) [4,9,11]
$d_{RNA_{cor}}$	degradation rate of RNA in cytoplasm	0.21 $h^{-1}$	[30]
$d_{DNA_{cor}}$	degradation rate of DNA in cytoplasm	0.03 $h^{-1}$	(0.016,0.2) [10,16,31,32]
$k_{DNA_{t}}$	transport rate of DNA from cytoplasm to nucleus	0.12 $h^{-1}$	[4,32]
$d_{DNA_{nuc}}$	degradation rate of DNA in nucleus	0.001 $h^{-1}$	[16]
$k_{int}$	integration rate	0.14 $h^{-1}$	(0.07,10) [4,33,34,35,36]
$d_{DNA_{int}}$	degradation rate of DNA integrated into chromosome	0.00002 $h^{-1}$	[37]
$TR_{cell}$	cell intrinsic rate of basal transcription	15 $h^{-1}$	[4,5]
$TR_{Tat}$	level of transcription induced by Tat transactivation	1500 $h^{-1}$	(1000,1500) [4,5]
$\theta_{Tat}$	threshold for half-maximal boosting of transcription by Tat	1000 molec.	[4,5], calibrated by [10]
$\theta_{Rev}$	threshold for half-maximal boosting on export of mRNAg and mRNAss by Rev	77,000 molec.	$\left(12,\;6\times 10^{5}\right)$, calibrated by [10]
β	inhibitory effect of Rev on the splicing rates implying their 1 / ( 1 − β ) -fold reduction at saturation level of Rev	0.9	(0.82,0.95) [4]
$k_{tp,RNA}$	transport rate of $RNA_{g}$ to cell membrane	2.8 $h^{-1}$	(1.44,43.2) [38]
$k_{eRNA_{i}}$	rate of $mRNA_{i}$ export from nucleus, i∈{g,ss}	2.3 $h^{-1}$	(2.1,4.6) [4,5,6]
$k_{eRNA_{ds}}$	rate of $mRNA_{ds}$ export from nucleus	4.6 $h^{-1}$	[4,5,6]
$k_{ssRNA_{g}}$	rate of splicing for full-length virus RNA	2.4 $h^{-1}$	(2,3) [4,5,6]
$k_{dsRNA_{ss}}$	rate of splicing for singly spliced virus RNA	2.4 $h^{-1}$	(2,3) [4,5,6]
$d_{RNA_{i}}$	degradation rate of $mRNA_{i}$, i∈{g,ss,ds}	$0.12\;h^{-1}$	(0.077,0.25) [4,5,6]
$d_{p,gp160}$	degradation rate of protein gp160	0.02 $h^{-1}$	[4]
$d_{p,j}$	degradation rate of protein *j*, j∈{Gag,Gag-Pol}	$0.09\;h^{-1}$	(0.05,0.39) [4,9]
$d_{p,Tat}$	degradation rate of Tat protein	$0.04\;h^{-1}$	(0.04,0.173) [4,5,6]
$d_{p,Rev}$	degradation rate of Rev protein	$0.07\;h^{-1}$	(0.04,0.173) [4,5,6]
$f_{ij}$	fraction of $mRNA_{i}$ coding $P_{j}$		
	i∈{g,ss,ds}, j∈{Gag,Gag-Pol,gp160,Tat,Rev}		
$f_{g,Gag-Pol}$	—	0.05	[4]
$f_{g,Gag}$	—	0.95	[4]
$f_{ss,gp160}$	—	0.64	[18]
$f_{ds,Tat}$	—	0.025	[4]
$f_{ds,Rev}$	—	0.2	(0.095,0.238) [4,5]
$k_{trans}$	rate of mRNA to proteins translation	524 proteins/mRNA/h	(50,1000) [4,5,6,39]
$d_{mem,Gag-Pol}$	degradation rate for the membrane anchored protein Gag-Pol	$0.004\;h^{-1}$	[40]
$d_{mem,Gag}$	degradation rate for the membrane anchored protein Gag	$0.004\;h^{-1}$	[40]
$d_{mem,gp160}$	degradation rate for membrane associated gp160 (Env)	$0.014\;h^{-1}$	[41]
$k_{tp,j}$	rate of protein $P_{j}$ transport to membrane, j∈Δ	$2.8\;h^{-1}$	(1.386,432) [4,38,42]
$k_{comb}$	incorporation rate of molecules into pre-virion complexes	8 $h^{-1}$	(6,12) [19,43]
$N_{RNA}$	number of viral RNA transcripts in a new virion	2	[13,19]
$N_{Gag}$	number of Gag molecules in a new virion	5000	(2500,5000) [19,21]
$N_{Gag-Pol}$	number of Gag-Pol molecules in a new virion	250	(125,250) [19,21]
$N_{gp160}$	number of gp160 molecules in a new virion	24	(12,105) [21,44]
$d_{comb}$	degradation rate of assembled pre-virion complex	0.52 $h^{-1}$	(0.33,1.25) [43]
$k_{bud}$	budding rate of new virions	2.0 $h^{-1}$	(1.3,4.5) [43]
$d_{bud}\equiv d$	degradation rate for budded immature viral like particle	0.38 $h^{-1}$	(0.38,1.5) [26]
	(= clearance rate of mature virions)		
$k_{mat}$	maturation rate	$2.4\;h^{-1}$	[8]

**Table 2 pathogens-09-00255-t002:** The most sensitive processes on which the net HIV-1 production depends.

Processes Having Negative Effect on *J*	*p*	$-p\frac{\mathrm{dJ}}{\mathrm{dp}}$	Processes Having Positive Effect on *J*	*p*	$+p\frac{\mathrm{dJ}}{\mathrm{dp}}$
Gag contribution to virion assembly	$k_{comb}N_{Gag}$	1810	Transcription induced by Tat	$TR_{Tat}$	1971
Degradation of free and mature virions	*d*	1721	Translation of Gag molecules	$k_{trans}f_{g,Gag}$	1811
Transport of genomic mRNA to membrane	$k_{tp,RNA}$	1697	Assembly of pre-virion complexes	$k_{comb}$	1810
Degradation of RNA during RT	$d_{RNA_{cor}}$	602	Transport of proviral DNA to nucleus	$k_{DNA_{t}}$	938
Degradation of assembled complexes	$d_{comb}$	364	Inhibitory effect of Rev on splicing rates	β	895
Degradation of DNA during RT	$d_{DNA_{cor}}$	268	Reverse transcription	$k_{RT}$	781
Splicing of full-length genomic RNA	$k_{ssRNA_{g}}$	262	Integration of proviral DNA	$k_{int}$	712
Degradation of budded immature particles	$d_{bud}$	242	Export of full-length genomic RNA	$k_{eRNA_{g}}$	415
Degradation of genomic mRNA	$d_{RNA_{g}}$	161	Budding of immature particles	$k_{bud}$	412
Tolerance of mRNA export and	$\theta_{Rev}$	118	Maturation of budded particles	$k_{mat}$	285
splicing to Rev-mediated regulation			Binding of virions to the cell membrane	$k_{bind}$	250
			Fusion of virions with the cell	$k_{fuse}$	166
			Translation of Rev molecules	$k_{trans}f_{ds,Rev}$	118
			Splicing of singly spliced RNA	$k_{dsRNA_{ss}}$	104

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
