# Peer review of "Modeling of the HIV-1 Life Cycle in Productively Infected Cells to Predict Novel Therapeutic Targets"

_pathogens, 2020, doi:10.3390/pathogens9040255_

Round 1

Reviewer 1 Report

Author's mathematical model describes the biochemical reactions
underlying the replication life cycle of HIV-1 in the activated target
cells. This study shows the two most critical stages in HIV-1 replication Gag
and Tat-Rev regulation of transcription and mRNA that can be targeted
to control virus production effectively.

Since the number of possible drug combinations and doses is too high
to test each in clinical trials, it is imperative to have models for
drug efficacy that can be used to make informed decisions about design
and administering therapy.

Author's models can provide essential insights into infection and can
be used to characterize the response of HIV to antiretroviral therapy.

Author Response

We thank the Reviewers for insightful comments and the thorough work on our manuscript.

Reviewer 1 stated the relevance of our mathematical model of intracellular HIV replication for giving insights into HIV infection and its control. There were no additional suggestions for modifying the manuscript.

Reviewer 2 Report

The manuscript by Shcherbatova et al. describe a mathematical model taking into account the different biochemical reactions of HIV-1 replication in target cells, calibrated using available information on the kinetics of various stages of HIV-1 replication. Using this model, authors identified HIV-1 assembly depending on Gag and Tat-Rev regulation of transcription and mRNA distribution as two of the most critical stages in HIV-1 replication in terms of  virion production. Given that these processes are not covered by current antiretrovirals, authors propose they can represent novel targets for antiviral therapy. 

Overall, the manuscript is surely of interest, given that the proposed model can aid in the identification of new, relevant therapeutic targets.

I have only minor comments, mainly regarding the discussion and the English language, which should be revised throughout the text. I report below the former, while the latter are indicated in the attached version of the manuscript through PDF comments and corrections.

  • I think that, given that the authors propose Gag and Tat-Rev as new potential targets, they should also briefly discuss the previous studies proposing their inhibition as antiviral strategies and the difficulties encountered. Can this model add some further insights on their development?
  • Does the model take into account viral load and CD4+ T cell count variation during infection course?

Author Response

We thank the Reviewers for insightful comments and the thorough work on our manuscript.

Reviewer 2 stated that the manuscript is surely of interest, given that the proposed model can aid in the identification of new, relevant therapeutic targets.

(1) I have only minor comments, mainly regarding the discussion and the English language, which should be revised throughout the text. I report below the former, while the latter are indicated in the attached version of the manuscript through PDF comments and corrections.

We have incorporated all the suggested corrections concerning the English language. Regarding the capitalization of protein names, we followed the notation adopted in Cold Spring Harbor Laboratory Press, Nature.

(2) I think that, given that the authors propose Gag and Tat-Rev as new potential targets, they should also briefly discuss the previous studies proposing their inhibition as antiviral strategies and the difficulties encountered. Can this model add some further insights on their development?

Indeed, there are numerals studies that propose more HIV inhibitors including Gag and Tat however they never made it into an antiviral drug. To mention this, we have now added 2 references that address these issues, and modified the sentence in line 177. It reads:” These processes while being considered as interesting antiviral targets (i.e. [47,48]) are not targeted by the current antiretroviral treatments”.

[47] Thenin-Houssier S, Valente ST. HIV-1 Capsid Inhibitors as Antiretroviral Agents. Curr HIV Res. 2016;14(3):270-82. https://doi.org/10.2174/1570162X14999160224103555

[48] Shin Y, Kim HG, Park CM, Choi MS, Kim DE, Choi BS, Kim K, Yoon CH. Identification of novel compounds against Tat-mediated human immunodeficiency virus-1 transcription by high-throughput functional screening assay. Biochem Biophys Res Commun. 2020 Mar 5;523(2):368-374. doi: 10.1016/j.bbrc.2019.12.029

(3) Does the model take into account viral load and CD4+ T cell count variation during infection course?

Our model is describing only the intracellular replication cycle of HIV but not HIV infection within a host organism. Only for the latter case, CD4 counts and viral loads would be an issue. This is planned however once we incorporate our present model into a multi-scale model to describe an HIV infection at the level of infected cells and an organism as a whole.

Reviewer 3 Report

The authors describe the intracellular life cycle of HIV-1, and introduce detailed mathematical equations for each step to develop a model of HIV-1 replication. The model is then used to identify which stages of the cycle are optimal for targeting via antiviral drugs. The overview of the life cycle and explanation of what each component of the equations means, was thorough while still simple and helpful. The figures were also a useful guide.

I only have minor comments, and would mainly suggest editing for clarity.

For example:

Line 30: “the set of control impacts”, meaning is unclear.

Line 34: “CD8 T cells in terms of their frequency and spatial dynamics” is also a little unclear. Does this mean quantity/concentration of the cells and where they are?

Line 37: suggest deleting “until today” or revise

Line 91: “split” instead of “splitted”?

Line 177: is this sentence meant to suggest these process are NOT considered in treatments yet, or they are considered and yet there’s a caveat to that?

Other comments:

-line 2: Add “Virus” after immunodeficiency in defining HIV

-line 23: define AIDS

-Given the extensive discussion of the genome and replication, it might be worth adding a sentence to the introduction to describe the positive sense, single stranded nature of the genome (for readers outside the field of virology)

Section 2.1.2: perhaps identify the “proviral” DNA upon first description of it being generated rather than waiting until after the figure to introduce to term.

-Can a little bit more information be provided about the limitations of the model? Especially given the model relied on already available data regarding kinetics, but the authors mention that existing models and information about intracellular kinetics are also limited.

-Are there other uses for the model? For example, could it be used to tell us anything about mutation rates or quasispecies?

Author Response

We thank the Reviewers for insightful comments and the thorough work on our manuscript.

Reviewer 3 found our manuscript interesting and had some minor comments that we take here in consideration as detailed below.

(1) Line 30: “the set of control impacts”, meaning is unclear.

We have now modified the phrase that now reads:

“The interventions and conditions that need to be explored in various combinations include ….”. 

(2) Line 34: “CD8 T cells in terms of their frequency and spatial dynamics” is also a little unclear. Does this mean quantity/concentration of the cells and where they are?

To be more precise, the phrase now reads:

“The requirements for cytotoxic CD8 T cells in terms of their frequency and spatial dynamics within lymphoid tissues have been examined in another study…”.

(3) Line 37: suggest deleting “until today” or revise

Since there are still ongoing efforts to develop new anti-HIV drugs, we prefer to leave the expression “until today” because it stresses the temporal aspect of the overall drug development process.

(4) Line 91: “split” instead of “splitted”?

We now changed to the correct form “split”.

(5) Line 177: is this sentence meant to suggest these process are NOT considered in treatments yet, or they are considered and yet there’s a caveat to that?

Indeed, the missing “not” escaped our attention. The phrase was corrected. This point was also addressed by Reviewer 2.

(6) -line 2: Add “Virus” after immunodeficiency in defining HIV

Done.

(7) -line 23: define AIDS

We have now modified the sentence to read:

“Intensive research on HIV and its induced acquired immunodeficiency syndrome (AIDS) for more ….”

(8) -Given the extensive discussion of the genome and replication, it might be worth adding a sentence to the introduction to describe the positive sense, single stranded nature of the genome (for readers outside the field of virology)

The nature of the viral genome as 2 single stranded RNA molecules was pointed out in the description of the HIV replication steps, see line 101:

“…a double-stranded DNA is created from two single-stranded RNA genomes”.

(9) -Section 2.1.2: perhaps identify the “proviral” DNA upon first description of it being generated rather than waiting until after the figure to introduce to term.

The proviral DNA form of HIV is already mentioned in line 60 where we describe other mathematical models that are in the literature. It reads

“… to integration of proviral DNA into the host genome.”

(10) -Can a little bit more information be provided about the limitations of the model? Especially given the model relied on already available data regarding kinetics, but the authors mention that existing models and information about intracellular kinetics are also limited. Are there other uses for the model? For example, could it be used to tell us anything about mutation rates or quasispecies?

Thank you for this suggestion and question about error rates and HIV quasispecies. We have now modified the end of the discussion to explain these. The text now reads:

“The current version of our model has the limitation that is does not consider the discrete nature and stochasticity of the biochemistry of the infection. This is inherent to the model set up as we considered a deterministic description for it. From the biological point of view, the model lacks a description of intracellular antiviral defense mechanisms and co-factors that facilitate virus propagation. Likewise, we did not incorporate error-prone virus replication that would lead to quasispecies distributions. All these factors can be incorporated to address specific issues that may arise including optimization of antiviral drug combinations and immune-based therapies. Further extensions of the model will require the analysis of stochastic effects to understand the link between the variability across the biochemical parameter values and the heterogeneity of virus production including the response to therapeutic perturbations.”